

# CNN-GMM approach to identifying data distribution shifts in forgeries caused by noise: a step towards resolving the deepfake problem

Roaa Mohamed Alnafea, Liyth Nissirat and Aida Al-Samawi

College of Computer Sciences and Information Technology, Department of Computer Networks, King Faisal University, Al-Ahsa, Saudi Arabia

## ABSTRACT

Recently, there have been notable advancements in video editing software. These advancements have allowed novices or those without access to advanced computer technology to generate videos that are visually indistinguishable to the human eye from real ones to the human observer. Therefore, the application of deepfake technology has the potential to expand the scope of identity theft, which poses a significant risk and a formidable challenge to global security. The development of an effective approach for detecting fake videos is necessary. Here, we introduce a novel methodology that employs a convolutional neural network (CNN) and Gaussian mixture model (GMM) to effectively differentiate between fake and real images or videos. The proposed methodology presents a novel CNN-GMM architecture in which the fully connected (FC) layer in the CNN is replaced with a customized Gaussian mixture model (GMM) fully connected layer. The GMM layer utilizes a weighted set of Gaussian probability density functions (PDFs) to represent the distribution of data frequencies in both real and fake images. This representation indicates there is a shift in the distribution of the manipulated images due to added noise. The CNN-GMM model demonstrates the ability to accurately identify variations resulting from different types of deepfakes within the probability distribution. It achieves a high level of classification accuracy, reaching up to 100% in training accuracy and up to 96% in validation accuracy. Notwithstanding the ratio of the genuine class to the counterfeit class being 16.6% to 83.4%, the CNN-GMM model exhibited high-performance metrics in terms of recall, accuracy, and F-score when classifying the least genuine class.

# INTRODUCTION

The past few decades have seen the proliferation of deepfakes, which are fabricated media content generated using deep learning techniques, that have emerged as a significant concern. Deepfake technology facilitates the generation of visual content, including images and videos, that possess a high degree of realism, making them indistinguishable from genuine counterparts when assessed using conventional detection techniques. The deepfake technique enables the overlaying of a face image of a target individual onto a

Corresponding author
Aida Al-Samawi,
aalsamawi@kfu.edu.sa

video or image of a source individual, thereby creating deceptive perceptions of the target person's presence and actions that lack veracity. Even though deepfake technology has been used in positive applications, such as its utilization in generating digital avatars, Snapchat filters, visual effects, or reconstructing missing segments in a film episode without necessitating reshoots (*Marr, 2019*), it also presents significant risks. The potential negative consequences of this phenomenon extend beyond the realms of personal harm and organizational blackmail. It has the potential to serve as a shield for individuals engaged in illegal activities by refusing to acknowledge video recordings that could be used as incriminating evidence against them. Therefore, the expeditious development of deepfake technology may significantly impact the value and authenticity of video evidence used in legal proceedings (*Maras & Alexandrou, 2018*).

Contemporary society is characterized by the advent of digital transformation. The emergence of deepfakes has the potential to significantly expand avenues for perpetrating identity theft, thereby endangering governmental and banking transactions. Deepfake has the potential to generate fabricated satellite imagery of the Earth by incorporating non-existent objects with the intention of misleading military analysts. The utilization of deepfake techniques to generate fabricated videos featuring world leaders delivering counterfeit speeches presents a significant obstacle to global security. Consequently, the utilization of deepfakes has the potential to incite political or religious discord among nations. Moreover, the ability to generate deepfake images or videos has become accessible to a wide range of individuals due to the emergence of advanced tools and mobile applications such as DeepFaceLab (*GitHub, 2020*), Reface (*Reface, 2024*), and Zao (*Jooste, 2023*). These technological advancements possess significant capabilities that greatly facilitate the deepfake process, rendering it more efficient and expeditious. Hence, the task of identifying deepfakes has become progressively more difficult.

Deepfakes detection research brings up in addition moral challenges about weaponization, bias, privacy, freedom of speech, responsibility, and openness. Bias against specific demographic groups may be reinforced by biased data sets, in addition to privacy concerns resulting from the analysis of facial features. When attempting to identify deepfakes for malevolent intent, it is also critical to avoid inhibiting legitimate uses of deepfakes. Furthermore, researchers must think about the possible ramifications of their work and put safeguards in place against misuse because weapon detection technology has the potential to suppress dissent, discredit individuals, and manipulate public opinion. Transparency is necessary for public accountability and scrutiny to take place. It is vital to look into how the discovery of deepfakes impacts public confidence in the media and broadens public awareness through educational initiatives. A safer online environment can be achieved by addressing these ethical issues.

Another challenge in detecting deepfakes is that deepfake generators employ counter-forensic techniques to avoid detection. Current detection strategies are unable to keep up with deepfake generators because they rely on particular features and processing requirements. Convolutional neural network (CNN) methodologies exhibit a reliance on contextual factors, vulnerability to overfitting, and a dependence on extracting high-level semantics from visual data.

This study aims to present a resilient approach for identifying deepfake content through the development and training of a CNN-GMM model that replaces the fully connected layer in the convolutional neural network (CNN) with a custom Gaussian mixture model (GMM) layer. The Gaussian mixture model (GMM) was used in the advanced topology to depict the probability density function (PDF) of features. Hence, any alteration to the image, including the utilization of deepfake techniques, will cause a shift in probability towards the presence of noise, thereby facilitating the recognition of such alterations. The CNN-GMM model takes advantage of the CNN's ability to extract high-level features from images and the GMM's capability to model complex data distributions. The GMM layer generates a collection of features, each accompanied by a probability value that indicates its respective contribution and level of significance. Therefore, it compensates for the uncertainty in the data to help overcome the overlaps in FaceForensics++ categories and to enhance model generalization.

The subsequent sections of the article are organized in the following manner: the Related Works section presents a comprehensive examination of various deepfake detection techniques that are relevant to deepfake detection. The Methodology section delineates the methodology used in this study, encompassing the pre-processing procedures, the proposed topology, and the GMM layer. The Experimental Setup section presents the experimental configuration, encompassing the hardware specifications employed for model training, the training parameters utilized, and the optimization of these parameters to attain the utmost accuracy. The Results section presents the outcomes, encompassing the training and validation accuracy, as well as the testing results. In the Conclusions section, a summary and conclusion of the study are provided.

## Related works

Back in late 2017, Deep learning was first used by an anonymous Reddit user known as "deepfakes" to swap celebrities' faces into pornographic videos (*Mirsky & Lee, 2021*). Later, in 2018, an alleged deepfake video of former president Barak Obama speaking on the subject was released by BuzzFeed. The video was made using the Reddit user's software: FakeApp (*Fake, 2016*). Consequently, concerns were raised over the spread of misinformation on social media and its impact. After that, it became obvious how such technology may be misused when University of Washington academics published a deepfake video of President Barack Obama and then spread it online. The researchers had complete control over what was said in the video of President Obama. Imagine what may happen if dishonest actors passed off a profound phony message from a world leader as the real thing. The security of the entire globe may be at risk.

At the beginning of 2019, a video of Nancy Pelosi, a US Speaker was manipulated in a "shallow fake" way to make her appear as if she was slurring her words and was inebriated or confused. Shallow fakes mean that audio-visual manipulations are done with less expensive and more easily accessible software. It involves basic video editing techniques such as slowing down, speeding up, cropping, and selectively splicing unmodified existing shots together, which can change the entire context of the information.

Fake videos are rapidly spreading on social media and their impact on public opinion highlights the significance of deepfakes and the reasons for the huge interest in them, which have encouraged researchers to focus on methods of distinguishing between fake and real media. A number of these detection methods rely on specific weaknesses that can be found in deepfake images/video which are left by the deepfake generation technique, while others use deep learning, and some researchers employ ensemble models to provide a more robust method of detection.

***Classifiers based on certain artifacts.*** In deepfake detection, some of the methodologies rely on certain vulnerabilities in the existing deepfakes. For example, an audiovisual methodology is employed to identify inconsistencies between the visual movements of the lips and the corresponding speech in an auditory context (*Agarwal et al., 2020*). The classifier can identify deficiencies in the ability of the deepfakes to accurately imitate mouth movements, synchronize lip movements, and reproduce corresponding speech. An additional approach that may be used to identify deepfake videos is to detect the frequency of blinking, as humans typically blink approximately once every 2 to 10 s (*Bentivoglio et al., 1997*), with each blink lasting approximately half to a quarter of a second (*Bartoshuk & Schiffman, 1977*). Deepfake videos exhibit a distinct characteristic of minimal blinking among individuals, thereby enabling their differentiation from real videos (*Li, Chang & Lyu, 2019*). Another proposed approach for detecting deepfakes involves analyzing eye color variation in videos (*Matern, Riess & Stamminger, 2019*). This can be accomplished by first segmenting facial regions from the images and then identifying iris pixels to assess the corresponding eye colors. *Yang, Li & Lyu (2019)* observed that deepfakes are generated by integrating artificially generated facial regions into an authentic image, which consequently introduces inaccuracies that could be detected through inconsistent head poses. *Salvi et al. (2023)* using an image decomposition module and multi-level feature enhancement, proposes a network to detect deepfake videos by highlighting inconsistencies in illumination. *Zhu et al. (2024)* suggested time-aware neural networks to extract audio-visual features from the input video over time. In their study, the inconsistencies between and within both video and audio modalities are exploited to enhance the final detection performance. Moreover, *Guarnera, Giudice & Battiato (2020)* study concentrate on creating a new forensics trace detection technique by analyzing deepfakes of human faces and extracting local features using an Expectation Maximization algorithm. Classifiers that rely on specific features demonstrate a high level of accuracy in detection, however, these classifiers are based on features that are heavily reliant on their statistical model. These approaches often encounter failures when the creation methods are enhanced and the underlying hypotheses no longer hold (*Verdoliva, 2020*). In addition, these approaches are not able to overcome improvements made to the new GAN models to make them undetectable. Deepfake variations are often unfamiliar in real-world scenarios, necessitating the development of a model that can effectively generalize and identify previously unseen instances of forgery.

***Deep learning-based methods*** in which features can be directly learned from data like CNN have been used frequently in deepfake detection (*Verdoliva, 2020*). *Pan et al. (2020)* used two deep learning models: the Xception model, a deep CNN architecture, and

MobileNets architecture which is suitable for mobile vision apps. Eight models were trained based on these two architectures and the four manipulation techniques were used in the FaceForensics++ dataset. These include Deepfakes, Face2Face, FaceSwap, and NeuralTextures. The results showed high performance over these trained models with accuracy ranging from 91% to 98%.

However, *Hsu, Zhuang & Lee (2020)* proposed a method based on a pairwise strategy in which a real-fake image pair was created using the CelebA dataset through five different states of art GANs: Deep convolutional GAN (*Radford, 2016*), Wasserstein GAN (*Arjovsky, Chintala & Bottou, 2017*), WGAN with gradient penalty (*Gulrajani et al., 2017*), least squares GAN (*Mao et al., 2017*) and PGGAN (*Huang et al., 2017*). Then, a two-stream model was proposed to input fake and real pairs, and the proposed common fake feature network (CFFN) was trained using pairwise learning. By aggregating the cross-layer feature representations, CFFN enabled fake feature learning middle- and high-level discriminative fake features. The trained fake image detector was then able to detect the fake image produced by a new GAN, even if it was not included during the training phase.

*Zhao et al. (2021)* developed a multi-attentional deepfake detection network with spatial attention heads, textural feature enhancement block, and aggregated low-level and high-level semantic features. It also introduces regional independence loss and attention-guided data augmentation strategy to address learning difficulties. Moreover, *Ilyas, Javed & Malik (2023)* created a deep learning framework called AVFakeNet, which can identify deep fakes in both visual and audio modalities. It is a Dense Swin Transformer Net (DST-Net) that uses a customized Swin transformer module in the feature extraction block and dense layers in the input and output block. This framework effectively identifies deepfakes in videos by carefully investigating both the audio and visual streams. On the other hand, *Chen et al. (2023)* proposed a Secure DeepFake Detection Network (SecDFDNet) that can detect DeepFake faces without disclosing private input while preserving a high accuracy level.

***Model ensembling*** is widely used in the field of machine learning to enhance the performance of detection tasks and mitigate the generalization error. This technique involves the amalgamation of sub-models or base learners to construct an optimal perceptual model. *Bonettini et al. (2021)* proposed the utilization of CNN architectures, specifically EfficientNetB4, EfficientNetB4Att, EfficientNetB4ST, and EfficientNetB4AttST, in conjunction with an attention mechanism and a Siamese triplet training scheme to facilitate the extraction of deep features. The Deepfakestack framework, developed by *Rana & Sung (2020)*, integrated a collection of advanced deep learning classifiers and was trained using the FaceForencis++ dataset. The DeepfakeStack Classifier, which is a CNN-based classifier, was developed and integrated into a larger multi-headed neural network. This integration was done to attain the most effective amalgamation of predictions derived from each source base-learner. The accuracy rate of the proposed DFC model, a larger stacking ensemble neural network, was found to be 99.65%. Moreover, the ensemble hierarchical model proposed by *Silva et al. (2022)* incorporated human involvement in the detection process by utilizing detection networks that employed both standard and attention-based data augmentation techniques. Attention blocks were

utilized to assess facial regions, while human analysis of the frequency and statistical analyses of the region revealed by the explanation layer were used to ascertain the validity of the frame. The model demonstrated a commendable accuracy rate of 92.4% when evaluated on the DFDC dataset, which is known for its complexity. Furthermore, the model's performance remained consistent even when presented with previously unseen data.

CNN-based techniques have demonstrated promising outcomes; however, their effectiveness is contingent upon the specific context in which they are applied. These techniques are prone to overfitting and are reliant on the extraction of high-level semantics from images (*Hulzebosch, Ibrahimi & Worring, 2020*). Furthermore, they are susceptible to being deceived by adversarial examples. Based on a comprehensive review of relevant literature and existing methodologies for detecting deepfakes, Gaussian mixture modeling has not been extensively explored in the identification of deepfakes. To date, there has been no research that has examined the combination of the GMM and the CNN for deepfake detection. Therefore, the authors propose the integration of a GMM layer into a CNN architecture, resulting in the proposed topology referred to as CNN-GMM. The rationale behind integrating GMM into our model is to effectively capture the underlying probability density function (PDF) of the features. Hence, any alteration made to the image, including the application of deepfake techniques, would cause a shift in the probability distribution towards the noise component, thereby facilitating the detection of such modifications by the GMM.

## METHODOLOGY

This section provides a detailed description of our methodology, including an overview of the FaceForensics++ dataset, the pre-processing techniques used, the proposed topology, and an explanation of the developed GMM layer.

### Dataset overview

Selecting an appropriate forensics dataset is essential for developing robust deepfake detection models. Important selection factors include the kind of manipulation, the methods used, the quality and resolution of the videos, the size and diversity of the dataset, and more. These factors have led to the selection of the FaceForensics++ dataset by *Rossler et al. (2019)* which involved five distinct processing techniques, including the manipulation of identity and expression. With varying quality levels and a variety of samples in terms of gender, race, and appearance, the dataset was updated, and new processing techniques were added. The FaceForensics++ includes a total of 6,000 videos, including 1,000 real videos, 4,000 manipulated videos, and 1.8 million fake frames. The videos in this dataset were created using five distinct processing techniques, namely Deepfakes, Face2Face, FaceSwap, NeuralTextures, and Faceshifter. The FaceForensics++ dataset provides a comprehensive set of techniques, enabling the model to effectively generalize to unfamiliar data and complex deepfake techniques.

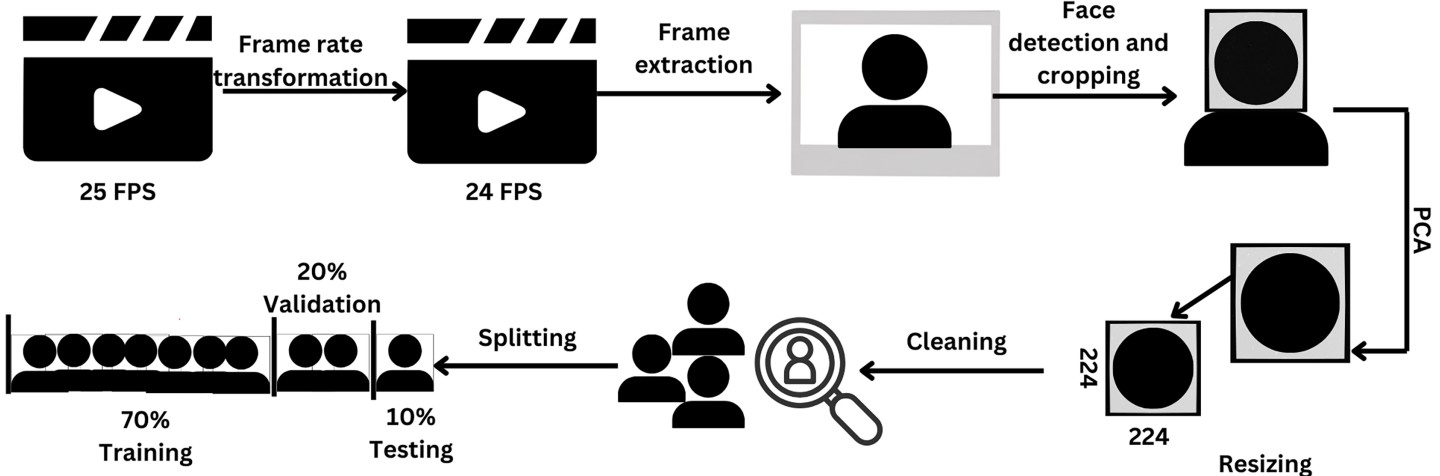

**Figure 1 Pipeline of preprocessing steps.** Figure created in Canva.

## Pre-processing

The primary objective of employing preprocessing techniques is to acquire precise, comprehensive, and coherent data within the dataset before inputting it into the deep learning model, thereby leading to improved classification performance (*Fan et al., 2021*).

In this work, the data underwent a series of transformations, including reduction, the application of principal component analysis (PCA), cleaning, partitioning, and augmentation. The preprocessing procedures employed in this study are depicted in Fig. 1.

The frame rate of the dataset videos was adjusted from 25 to 24 to ensure compatibility with the MATLAB platform version 2022a (MathWorks, Natick, MA, USA). Subsequently, the frame was extracted, and the face was detected through the Viola-Jones algorithm (*Viola & Jones, 2001*), which is the pioneering real-time face detection algorithm utilizing Haar-like features. The image was then resized to 224 × 224 pixels. Afterward, PCA was employed on the image to address the issue of high correlation among the red (R), green (G), and blue (B) matrices. PCA is a linear technique used for reducing the dimensionality of a dataset. It accomplishes this by transforming a set of correlated variables into a smaller set of uncorrelated variables, referred to as principal components (*Jolliffe & Cadima, 2016*). The objective of PCA is to retain as much of the original data's variability as possible during this transformation process. Following PCA, the resulting images undergo a manual inspection process to identify and remove any instances of improper detection.

A total of 70% of the data were allocated to the training set and 30% were used for validation purposes (*Nguyen et al., 2021*). Furthermore, the validation data was additionally subdivided, with 67% of the data assigned for validation and the remaining portion assigned as the test data.

Data augmentation techniques were used to enhance the generalization capabilities of the model to augment the training dataset, thereby increasing its size. The datasets were

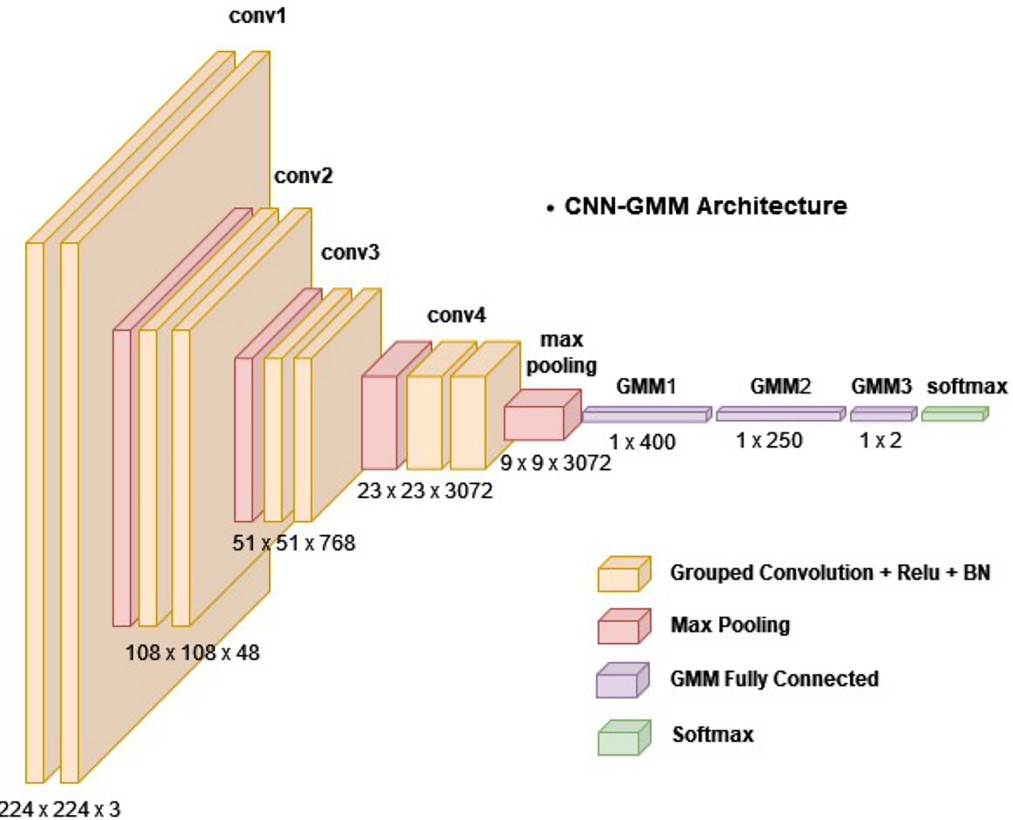

**Figure 2 The proposed model (CNN-GMM) architecture.**

enriched with a variety of effects that would enhance the validation (*Moradi, Berangi & Minaei, 2020*). The used dataset was augmented through the implementation of two distinct types of transformations:

- **Image reflection:** Vertical or horizontal random reflection in the left-right and top-bottom direction; each image is reflected with 50% probability.
- **Image translation:** The image is simply moved along the X or Y axes based on the given interval. A uniform continuous distribution within the specified interval [−4 4] is used to pick the vertical or horizontal translation distance.

## Topology

The depicted topology is presented in Fig. 2. The model was designed to process image input of dimensions $224 \times 224 \times 3$ pixels. This input was then subjected to four consecutive blocks of convolutional operations followed by pooling. A block is comprised of vertically arranged layers, with each layer fulfilling a distinct purpose. The CNN-GMM architecture is made up of multiple blocks, where each block is comprised of two grouped convolutions. Following each convolution operation, rectified linear unit (ReLU) activation and batch normalization were applied and a max pooling layer was employed. Then, a single fully

**Table 1  Parameters of each layer in CNN-GMM.**

| Layer name | Number of filters | Size of feature map | Size of kernel | Stride |
|---|---|---|---|---|
| 1. Grouped convolution 1 | 3 groups with 4 filters in each | 222 * 222 * 12 | [3 3] | [1 1] |
| ReLU + Batch normalization | | | | |
| 2. Grouped convolution 2 | 12 groups with 4 filters in each | 220 * 220 * 48 | [3 3] | [1 1] |
| ReLU + Batch normalization | | | | |
| Max pooling 1 | – | 108 * 108 * 48 | [5 5] | [2 2] |
| 3. Grouped convolution 3 | 48 groups with 4 filters in each | 106 * 106 * 192 | [3 3] | [1 1] |
| Relu + Batch normalization | | | | |
| 4. Grouped convolution 4 | 192 groups with 4 filters in each | 104 * 104 * 768 | [3 3] | [1 1] |
| ReLU + Batch normalization | | | | |
| Max pooling 2 | – | 51 * 51 * 768 | [3 3] | [2 2] |
| 5. Grouped convolution 5 | 786 groups with 2 filters in each | 49 * 49 * 1,536 | [3 3] | [1 1] |
| ReLU + Batch normalization | | | | |
| 6. Grouped convolution 6 | 1,536 groups with 2 filters in each | 47 * 47 * 3,072 | [3 3] | [1 1] |
| ReLU + Batch normalization | | | | |
| Max pooling 3 | – | 23 * 23 * 3,072 | [3 3] | [2 2] |
| 7. Grouped convolution 7 | 3,072 groups with 1 filter in each | 21 * 21 * 3,072 | [3 3] | [1 1] |
| ReLU + Batch normalization | | | | |
| 8. Grouped convolution 8 | 3,072 groups with 1 filter in each | 19 * 19 * 3,072 | [3 3] | [1 1] |
| ReLU + Batch normalization | | | | |
| Max pooling 4 | – | 9 * 9 * 3,072 | [3 3] | [2 2] |
| 9. Fully connected layer | – | 1 * 400 | – | – |
| Dropout (0.4) + Batch normalization | | | | |
| 10. GMM fully connected layer | – | 1 * 400 | – | – |
| Batch normalization | | | | |
| 11. GMM fully connected layer | – | 1 * 250 | – | – |
| Batch normalization | | | | |
| 12. GMM fully connected layer | – | 1 * 2 | – | – |
| SoftMax layer | | | | |
| Classification layer | | | | |

connected layer was employed after the convolutional blocks, followed by three developed GMM layers. The layer operations will be explained in detail in the next section. The first layer of the Gaussian mixture model (GMM) consisted of 400 neurons, the second layer contained 250 neurons, and the final layer of the GMM was composed of two neurons, as it is designed for binary classification.

There was a total of 38 layers in this network with 14 layers in the model, each of which possessed learnable weights. Among the model layers, nine were convolutions, one was fully connected, and three were GMM layers. In the CNN-GMM architecture, the convolutional layers employed grouped convolutions with [3 3] filters and a stride value of 1. The initial pooling layer had [5 5] filters with a stride of 2, while the subsequent pooling

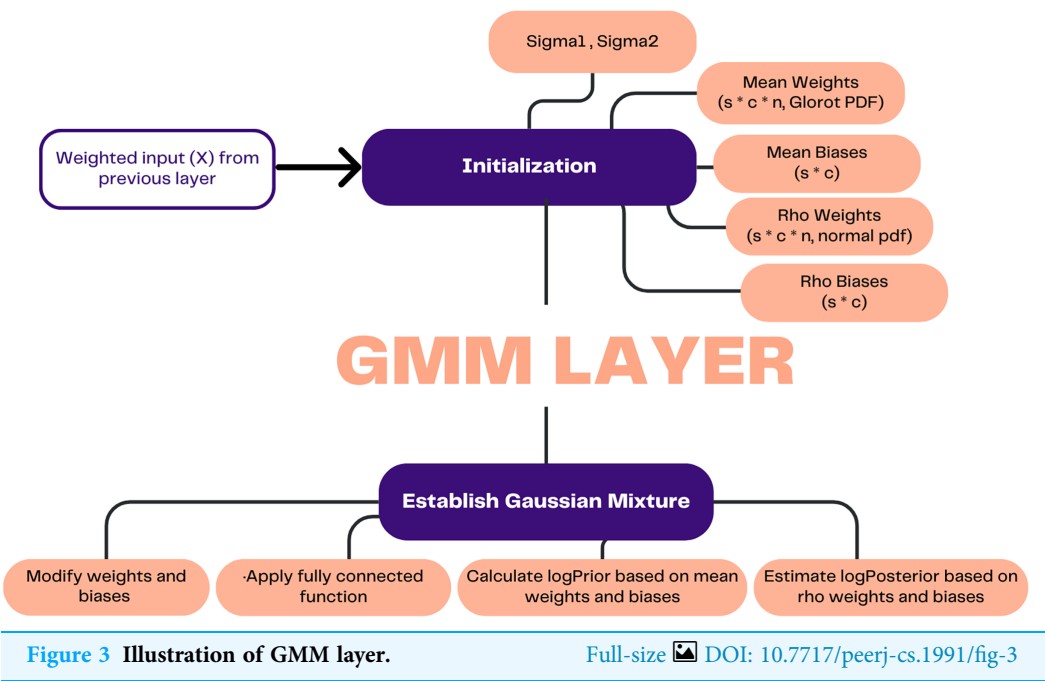

**Figure 3 Illustration of GMM layer.**

layers used [3 3] filters. Table 1 depicts the configuration of the layers and their respective parameters. The counting was restricted to layers that possess learnable weights.

## GMM layer

Weight uncertainty in the context of deep learning pertains to the extent of variability exhibited by a weight parameter, such that its values may be altered within a certain range without causing a substantial impact on the overall performance of the model. This aids in enhancing the generalization performance and precision of predictions on unfamiliar data. A GMM layer was developed to estimate weight uncertainty by considering the network weights as random variables. This enabled the neural network to acquire knowledge about the probability distribution of weights, thereby facilitating the estimation of prediction uncertainty.

The functionality of this layer was achieved through the utilization of a prior distribution, which serves to represent the inherent uncertainty about the weights associated with the layer. The prior distribution refers to a probability distribution that characterizes the weights of a layer before its training. During the training process, the prior distribution is updated by the layer, taking into consideration four distinct parameters. The parameters under consideration include rho weights, rho biases, mean weights, and mean biases. The posterior distribution represents the revised probability distribution after incorporating new information that was initially represented by the prior distribution. Predictions regarding the layer's output can be made using the posterior distribution.

Figure 3 demonstrates how the process in this layer commences by receiving the weighted input (X) from the preceding layer. Subsequently, the process involves

establishing initial values for the standard deviation of the weights (Sigma1) and the standard deviation of the biases (Sigma2), which serve as indicators of the variances within the mixture of Gaussian distributions that characterize the prior distribution. Furthermore, the initial values are assigned to the weights and biases corresponding to the mean and rho. The weights and biases, which can be modified through learning, are updated and subjected to a random epsilon. Subsequently, the learnable parameters are subjected to a fully connected function, wherein the input is multiplied by the weights matrix and subsequently augmented by the bias term.

The prior probability was computed, and the posterior probability was estimated based on the updated weights and biases. The calculation of prior probability involved the utilization of the natural logarithm and the Gaussian mixture model, which incorporates mean weights and mean biases. Equation (1) is utilized to estimate the weights and biases, while Eqs. (2) and (3) are employed for the computation of the prior probability.

$$logPDFWeights = scale\left(-0.5 \times \frac{\left(x_w^{n-1} - \mu_w^{n-1}\right)}{\sigma_1^{n-1}} - 0.5 \times log(2\pi) - log\left(\sigma_1^{n-1}\right)\right)$$
$$+ (1 - scale)\left(-0.5 \times \frac{\left(x_w^{n-1} - \mu_w^{n-1}\right)}{\sigma_2^{n-1}} - 0.5 \times log(2\pi) - log\left(\sigma_2^{n-1}\right)\right). \tag{1}$$

The above equation is repeated for bias, for weights, $x_w^{n-1}$ is the sampled weights, and for biases it is the sampled biases.

$$logMixturePrior = logPDFWeights + logPDFBias \tag{2}$$

$$logPrior = \sum_{i=1}^{number\ of\ wieght\ matrix\ elements} logMixturePrior_i. \tag{3}$$

In contrast, the estimation of the posterior probability relies on the utilization of updated rho weights and rho biases, employing the Glorot initialization technique (*Glorot & Bengio, 2010*). The Glorot initialization method ensures that the weights of a neural network are not excessively amplified or suppressed during training, thereby facilitating ease of training. This is achieved by independently sampling the weights from a uniform distribution. The subsequent equations provide a means of estimating the posterior probability. Equations (4), and (5) used to estimate the logPosterior weights and Eqs. (6) and (7) for logPosterior biases (*Blundell et al., 2015*):

$$logRho_{weights}^n = log\left(1 + e^{rho_{weights}^n}\right) \tag{4}$$

$$logProbabilityPosteriorWeights = \sum_{i=1}^{number\ of\ weight\ matrix\ elements} -0.5 \times \frac{\left(x_w^n - \mu_w^n\right)}{\left(logRho_{weights}^n\right)^2} \tag{5}$$
$$- 0.5 \times log(2\pi) - log\left(logRho_{weights}^n\right)$$

$$logRho_{bias}^n = log\left(1 + e^{rho_{bias}^n}\right). \tag{6}$$

**Table 2 Adam optimizer parameters.**

| Parameter | Epsilon | Mini batch size | Gradient decay factor | Initial learn rate | shuffle |
|-----------|---------|-----------------|------------------------|--------------------|---------|
| Value | 1e−8 | 80 | 0.9 | 0.001 | 'every-epoch' |

$$logProbabilityPosteriorBias = \sum_{i=1}^{number\ of\ bias\ matrix\ elements} -0.5 \times \frac{(x_b{}^n - \mu_b{}^n)}{(logRho_{bias}{}^n)^2} - 0.5 \qquad (7)$$
$$\times\ log(2\pi) - log(logRho_{bias}{}^n)$$

To estimate logPosterior:

$$logPosterior = logProbabilityPosteriorWeights + logProbabilityPosteriorBias. \qquad (8)$$

The trainable parameters, namely weights and biases are used to create uncertainty in the features. In contrast to a conventional fully connected layer that generates a static feature, a GMM yields a collection of features, each accompanied by a probability that indicates its respective contribution and significance. Consequently, the incorporation of compensation for data uncertainty will prove beneficial in effectively addressing the issue of category overlaps and biases.

CNN-GMM utilizes the GMM fully connected layer as a substitute for the conventional fully connected layer. It was not necessary to use the dropout after the GMM layer due to its inherent capability to mitigate the adverse effects of overfitting.

## Experimental setup

This section provides an overview of the experimental configuration, including the hardware specifications used for model training, the training parameters employed, and the optimization of these parameters to achieve the highest level of accuracy.

## Development platform

MATLAB (MathWorks, Natick, MA, USA) was used to conduct these experiments.

### Hardware specification

The model was trained on a PC with the following specifications:
- 11th generation Intel(R), Core (TM) i7-1165G7 @ 2.80 GHz.
- 16 GB RAM.

It took approximately 54 h to train the model using in the local CPU.

### Training options

The adaptive moment estimation (Adam) optimizer was used (Table 2) (*Kingma & Ba, 2014*) due to its computational efficiency, low memory requirements, and invariance to diagonal rescaling of gradients.

Epsilon is a very small number used to prevent any division by zero in the implementation while updating the variable when the gradient is almost zero. The default

value for epsilon in the Adam optimizer (1e−8) typically works well for most cases. A mini batch is a portion of the training set used to assess the gradient of the loss function and modify the weights. Smaller batches may converge more quickly than larger ones. Also, due to its high variance, a small batch size might have a considerable regularization effect (*Wilson & Martinez, 2003*). Considering this concept, and the constraints imposed by the limited RAM capacity of both the PC and AWS GPU, a modest mini-batch size of 80 was used. The default value of 0.9 was used for gradient moving average decay rate in the Adam optimizer. The learning rate controls the step size at each iteration of an optimization algorithm as it advances toward a minimum of a loss function. It reflects the rate at which a neural network model learns since it determines the extent to which newly acquired information overcomes old ones. Training may take a while if the learning rate is too low (*Kim, 2017*) The training process may diverge or produce less than ideal results if the learning rate is too high. The initial learning rate was set in this work at 0.001, which is the default for the Adam optimizer. To avoid dismissing the same data repeatedly, training data was shuffled before every epoch.

## Parameter optimization

The goal of parameter optimization is to find a set of parameters that maximizes the algorithm's performance on a specific problem instance (*Huang et al., 2019*). The parameters were manually adjusted to improve accuracy.

The mini-batch size was tuned from 300 to 80 to regularize and enhance training performance. Reducing the size of the minibatch affects the generalizability of the model (*Kandel & Castelli, 2020*) and enables more efficient training with limited memory resources. Furthermore, through repeated experiments to optimize the system, it was noticed that changing the initial learning rate from 0.0001 to 0.001 increased the accuracy of the proposed topologies by approximately 3%. Moreover, the FC layer neuron should be proportional to the output vector from the previous layer. If a high number of neurons is used, it would result in an unmanageably large network, and if a small number is used, this will negatively affect the performance of the network. Therefore, FC neurons were adjusted until a value that balances performance and network size was reached. Further, increasing the epochs number gives the network enough time to be trained as the algorithm is exposed to the same data multiple times, which leads to a higher accuracy. Fifty epochs were sufficient for the proposed topology to be learned without overfitting.

## RESULTS

### Training results

The CNN-GMM model was trained for approximately 54 h on a personal computer CPU using the FaceForensics++ dataset. The model achieved a training accuracy of 100% and a validation accuracy of 96%, as depicted in Fig. 4. The training process is shown with a blue solid line representing training and a dashed red line representing smooth training. The red solid line in the figure represents the accuracy of validation. Figure 5 illustrates a loss function that exhibits a gradual decrease from a value of 3 to approximately 0.5.

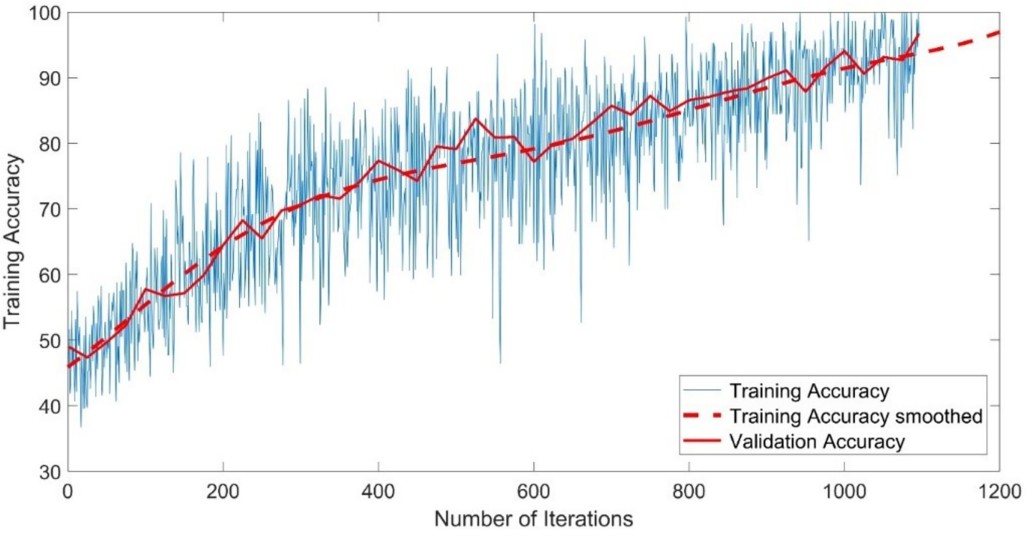

**Figure 4 Training accuracy of the CNN-GMM model.**

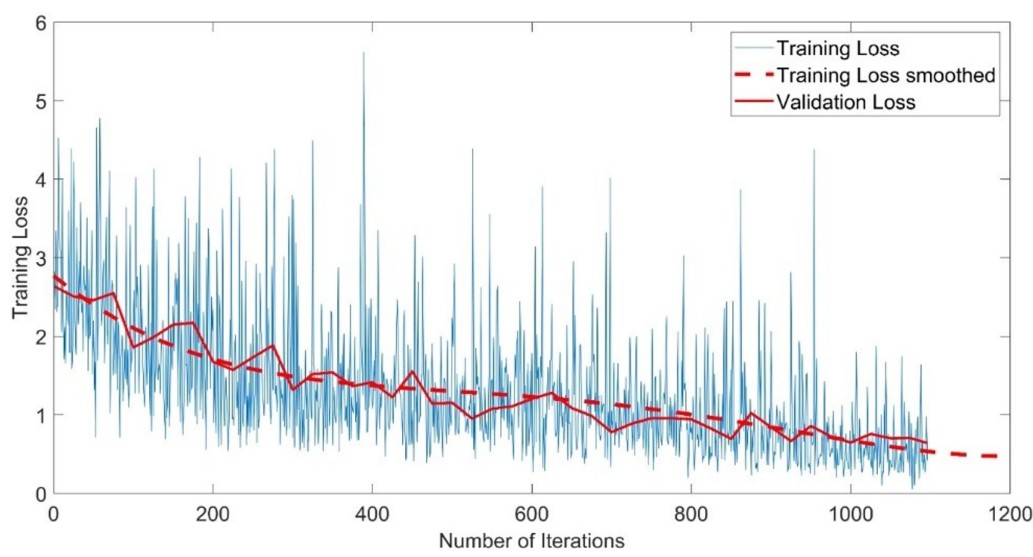

**Figure 5 Training loss of the CNN-GMM model.**

## Evaluating the model using the test set

The system underwent testing by classifying a test set that had been previously isolated from the FaceForensics++ dataset. Consequently, this test set was regarded as new data for the system. The recorded classification accuracy of the test was 96.2%.

This work comprises two distinct classes, namely Real (R) and Fake (F). As depicted in the first table of Fig. 6, the True Fake (TF) values, representing the correctly classified fake samples, account for 465 out of the total 484 fake samples (96.1%). The category of False Fakes (FF) encompasses 19 instances out of a total of 484 counterfeit samples, resulting in an erroneous classification rate of 3.9%. Out of a total of 97 samples, 94 (96.9%) True Real (TR) samples accurately represent the real values, while 3 (3.1%) False Real (FR) samples inaccurately represent the real values.

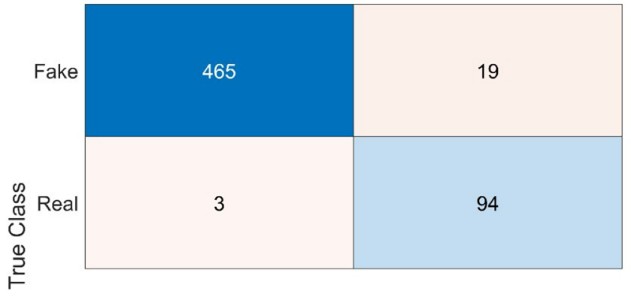
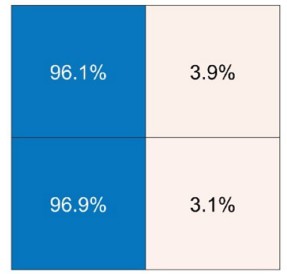
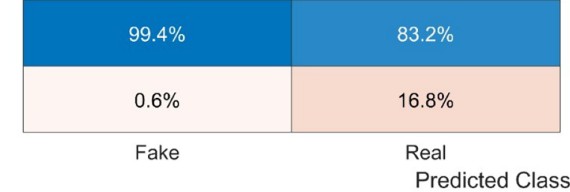

**Figure 6** **Confusion matrix of the testing results.**   

**Table 3  Confusion parameters of the testing result.**

|  | Accuracy | True positive/Recall/ SENSITIVITY | Predicted positive/ PRECISION | Actual negative/ SPECIFICITY | False positive/ fall-out | F-score |
|---|---|---|---|---|---|---|
| Fake | 96.2 | 99.4 | 96.07 | 83.2 | 16.8 | 96.63157895 |
| Real | 96.2 | 83.2 | 96.9 | 99.4 | 0.6 | 84.90566038 |

    The confusion parameters of this model are shown in Table 3. The test results demonstrate that the system exhibits strong generalization capabilities and effectively applies learned knowledge to new data without solely relying on memorization or overfitting to the trained data. The classification accuracy of the test data was 96.2%, which closely aligns with the training accuracy of 96%. Moreover, to determine the proportion of accurate true classifications within each class, the recall or true positive rate was computed. The recall value for the fake class (RecallFake) was 0.994 with a confidence level of 99.4% indicating a high probability of correctly classifying a fake video as fake. In contrast, the recall value for the real class (RecallReal) was 0.832 with a confidence level of 83.2%, indicating a lower probability of correctly classifying a real video as real. The precision, also known as the predicted positive value, was determined by dividing the count of accurate predictions by the sum of false predictions and accurate predictions. The value of PrecisionFake was 96.07, while the value of PrecisionReal was 96.9. The concept of specificity pertains to the model's capacity to accurately detect and classify negative outcomes. The level of specificity exhibited by the model in the Fake class was 83.2%, while in the Real class, it reached 99.4%. The calculation of the false positive rate (FPR) or fall-out involves dividing the count of negative events that have been incorrectly classified as positive by the total count of negative events. The false positive rate for the fake condition was 16.8%, while the false positive rate for the real condition was 0.6%. The F score, which quantifies the accuracy of the model, is calculated by taking the harmonic mean of the
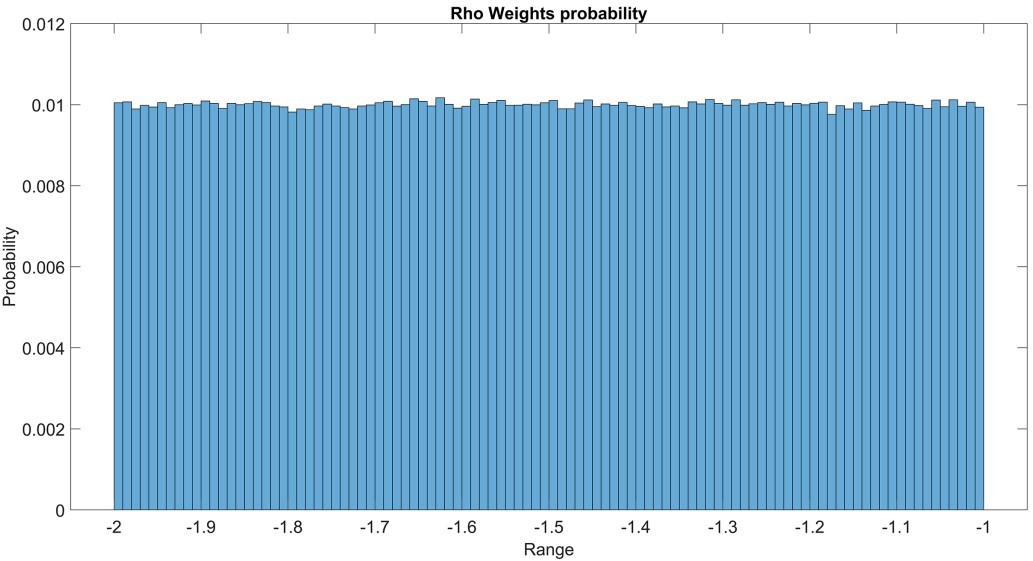

**Figure 7  Rho weights probability in the first GMM layer.**

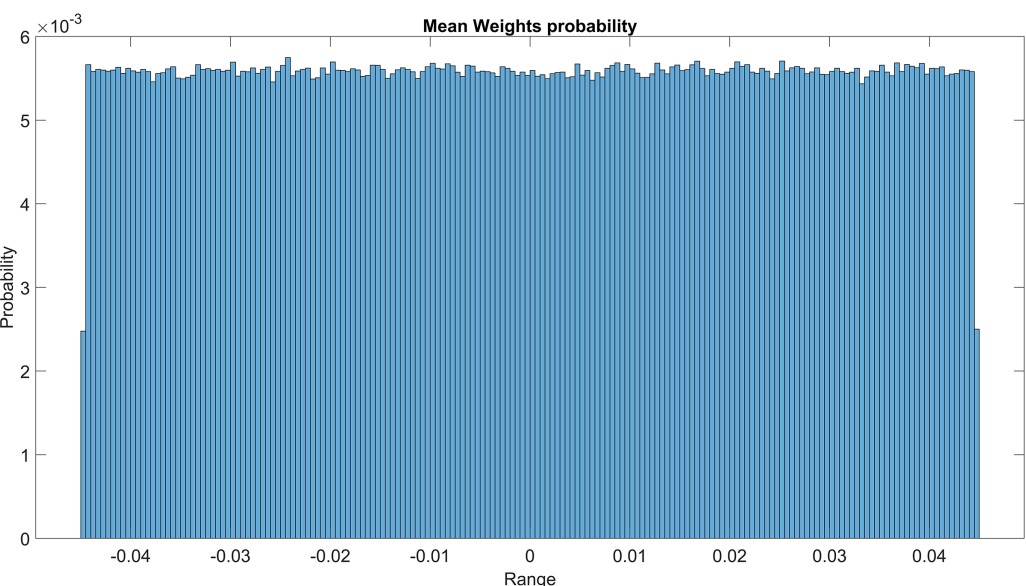

**Figure 8  Mean weights probability of the first GMM layer.**

precision and recall. In this case, the F score for the Fake classification was 98.1 and 89.5 for the Real class. As can be observed, the Fake class's precision, recall, and F score were all higher than those of the Real class, indicating that the system was biased in favor of the Fake class because of its size.

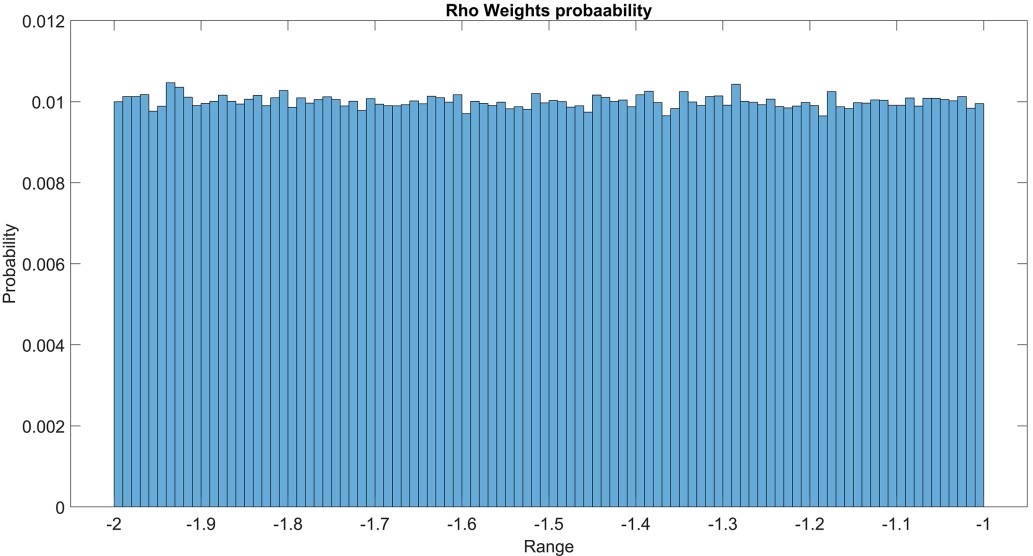

**Figure 9** Rho weight probability of the second GMM layer.

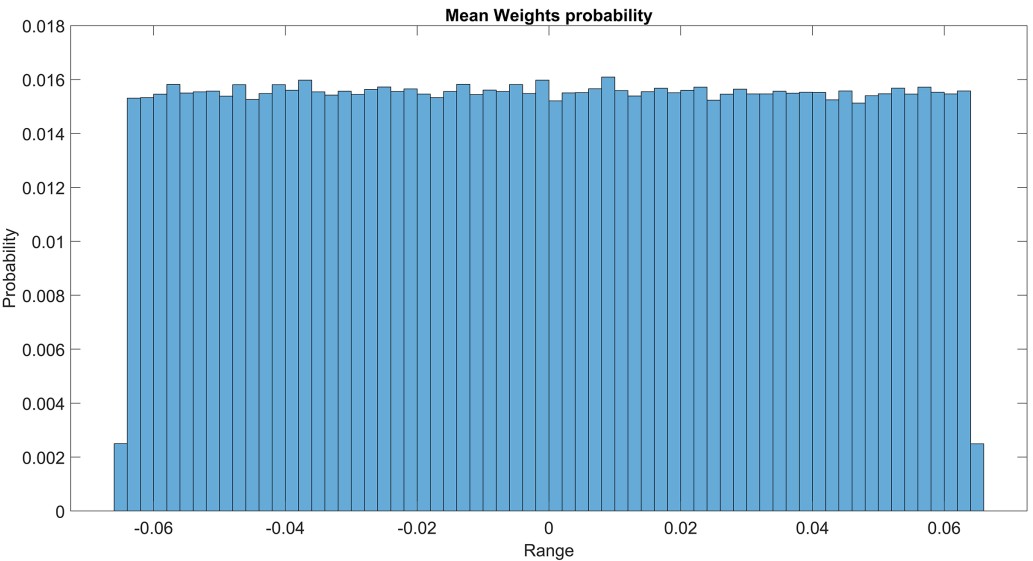

**Figure 10** Mean weight probability of the second GMM layer.

## Distribution of GMM layers

The GMM layer was designed with learnable rho weight and mean weight parameters. Figure 7 illustrates the distribution of rho weight values for the initial GMM layer. The weight values in question exhibit a uniform distribution, with an approximate probability of 0.01 assigned to each value. The weight values in the model are bounded within the range of −2 to 1, representing the weight uncertainties or the permissible range of values

that a weight can assume without causing a substantial impact on the model's performance. The imposition of a uniform weight distribution compels the neural network to allocate equal attention to the features extracted from the activation maps of preceding layers. This enhances the model's capacity for generalization and mitigates its bias. Figure 8 illustrates the probability distribution of the mean weight values within the first GMM layer, spanning a range of −0.04 to 0.04, and following a uniform distribution. Figure 9 illustrates the distribution of rho weights in the second GMM layer, spanning from −2 to 1. Figure 10 depicts a uniform distribution of mean weights in the same GMM layer, ranging from −0.06 to 0.06. It is worth noting that this range is slightly wider compared to the mean weights observed in the preceding GMM layer.

## CONCLUSIONS

Tools for manipulating videos have become increasingly more sophisticated, thereby facilitating the production of fake videos that are virtually indistinguishable from legitimate ones. These circumstances give rise to the possibility of exploiting deepfakes for nefarious objectives such as identity theft and political manipulation. Great efforts are being made by both creators and detectors of deepfakes, which increases the difficulty of producing undetectable deepfakes. Nevertheless, creators are continuously discovering novel methods to evade detection. The present study introduces an innovative deep learning model, namely the CNN-GMM model, for deepfake detection. The FaceForensics ++ database, encompassing five distinct manipulation methods, was used to train the CNN-GMM. The videos within this dataset have undergone a series of preprocessing steps, including transformation, reduction, PCA application, cleaning, splitting, and subsequent augmentation. The CNN-GMM model demonstrated a notable capability in identifying uncertainties arising from various categories of deepfakes within the probability distribution, achieving a commendable level of accuracy in classification. Specifically, the model attained a training accuracy of 100% and a validation accuracy of 96%. The CNN-GMM model demonstrated effectiveness in addressing the issue of imbalanced class distribution through rescaling. It also exhibited a high level of classification performance during testing, increasing its potential for generalizability. However, it may not be immune to biases present in the FaceForensics++ dataset such as demographic imbalance, a high proportion of male faces, limited representation of faces of European descent, age bias, quality disparity, and context bias. Consequently, to overcome this bias, it is suggested to train the model on more datasets in the future and feed it random samples from recent deepfake generation tools. In addition, lightweight models should be developed for real-time detection on resource-constrained devices, such as smartphones and tablets. The constant change in the landscape of deepfake detection necessitates scalable and resilient models. Existing detection methods are susceptible to new developments, necessitating robustness and scalability enhancements.

## ACKNOWLEDGEMENTS

We extend our thanks to the creators of the FaceForensics++ dataset for providing an invaluable resource that has significantly propelled our research.

### Funding
This work was funded by the Deanship of Scientific Research, Vice Presidency for Graduate Studies and Scientific Research, King Faisal University, Saudi Arabia (Project No. GRANT4,463). The funders had no role in study design, data collection and analysis, decision to publish, or preparation of the manuscript.

### Grant Disclosures
The following grant information was disclosed by the authors:
Deanship of Scientific Research, Vice Presidency for Graduate Studies and Scientific Research, King Faisal University, Saudi Arabia: GRANT4,463.

### Competing Interests
The authors declare that they have no competing interests.

### Author Contributions
- Roaa Mohamed Alnafea conceived and designed the experiments, performed the experiments, analyzed the data, performed the computation work, prepared figures and/or tables, authored or reviewed drafts of the article, and approved the final draft.
- Liyth Nissirat conceived and designed the experiments, performed the experiments, analyzed the data, performed the computation work, prepared figures and/or tables, authored or reviewed drafts of the article, and approved the final draft.
- Aida Al-Samawi conceived and designed the experiments, performed the experiments, analyzed the data, prepared figures and/or tables, authored or reviewed drafts of the article, and approved the final draft.

### Data Availability
The dataset is available at GitHub: https://github.com/ondyari/FaceForensics (Andreas Rössler, Davide Cozzolino, Luisa Verdoliva, Christian Riess, Justus Thies, Matthias Nießner).

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
