# Peer review of "CNN-GMM approach to identifying data distribution shifts in forgeries caused by noise: a step towards resolving the deepfake problem"

_PeerJ Computer Science, doi:10.7717/peerj-cs.1991_

## Round 0.1 · original submission · Major Revisions

Reviewers find merit to your paper, however, they recommended a major revision. You are required to address all the comments and suggestions of the reviewers and resubmit a revised manuscript. The revised manuscript will be subject to a 2nd round review. Good luck.

Reviewer 1 ·

Basic reporting

The introductory section provides a general overview of the Deepfake problem, its growing awaeness, and challenges related to security and privacy. However, it might benefit from a more detailed historical context and specific examples illustrating the evolution and current state of deepfake problem. This would enhance the readers' understanding of the topic and its relevance. Several new research studies should be included in the references for the topic. Authors must enhance the topics included in the recently proposed models by several researchers. The general English of the paper is not appropriate as per the journal ethics and guidelines. The paper is not thorougly checked for English editing and proof reading.

Experimental design

The research objectives appears to be well-defined and relevant, focusing on addressing deepfake detection method. The methods described, including the use of the Gaussian functions, seem appropriate for the research objectives. However, there is room for more explicit detailing of the procedures and protocols followed in the study to enable replication and assessment by other researchers.

Several questions like below should be answered by the authors to enhance the level of the article to the journal level as:
Why did the authors selected the given functions only?
Why did the authors used probablistic method only?
Authors claim that they developed a deepfake detection model, however, the methods used are not investigated properly. There lacks a proper justification for the availability of exisiting methods for the same and their failure reasons.
Why is the deepfake detector modelling make use of neural networks only??
The manuscript lacks the answer to the questions above.
Adding the reasons will add a lot of value to the article.
A training accuracy of 100% seems unrealistic. The author might have investigated a small domain of the dataset values for the problem, which might refer to very poor model results and findings.

Validity of the findings

The findings, including the proposed model’s high accuracy in deepfake detection, are promising but seems non realistic. However, the article might benefit from a more rigorous statistical analysis of the results. Additionally, comparisons with existing models or systems in similar domains could provide a clearer understanding of the impact and novelty of the study.
Several questions should be answered as below:
How is the training data trained for the deepfake detection? The value of probablistic Posterior Weights are determined by what factors and why?
The formula presented in the study are not represented properly. Indeed, Probability Posterier Weights are mispelled as "Propability" in the complete mathematical modelling.

Additional comments

Various components like the deepfake detection and neural network are discussed with technical details. However, a more comprehensive explanation of how these components interact and complement each other in the proposed system would be beneficial. Also, elaborating on the specific innovations or improvements each component brings to the deepfake detection field would add value. The article covers detection as critical aspects of the system. Further clarification on how these processes are specifically tailored or optimized for the deepfake context could enhance the article's relevance and applicability. Some references in the manuscript are not cited properly. It seems that the formatting of the article needs some modifications to make it acceptable as per the journal guidelines. Thus in general the article lacks value and proper reasoning to proceed for publication. The authors are requested to update the article as per the journal guidelines and proper reasoning.

·

Basic reporting

1- The article mostly uses clear and professional English but could benefit from proofreading for minor grammatical errors.
2- While it provides adequate literature references, some key studies in deepfake detection seem to be missing. Adding these would give a more comprehensive background.
3-The structure is professional, but some figures and tables could be more clearly labeled. Sharing raw data for transparency would also enhance the paper's credibility. Formal results are well-defined, but including more detailed proofs and term definitions would increase clarity.

Experimental design

The research aligns with the journal's scope and addresses a meaningful question. However, the methods section could be more detailed to facilitate replication by other researchers. Including more diverse datasets and explaining the selection criteria for these datasets would strengthen the experimental design. Additionally, discussing the ethical considerations of deepfake detection research would address important concerns in this rapidly evolving field.

Validity of the findings

The findings are relevant and robust but lack a comprehensive statistical analysis. Enhancing the statistical robustness and providing more control conditions would make the findings more convincing. The conclusions are well-connected to the research question, but they could be more cautiously stated, with limitations and potential biases acknowledged. Providing all underlying data would aid in verifying the results and increasing the paper's transparency.

Additional comments

the work could focus on the scalability of the CNN-GMM model and its adaptability to different deepfake technologies. Additionally, exploring the potential societal impacts and ethical implications of deepfake detection would provide a more holistic view of the research. Collaboration with experts in digital ethics could enrich this discussion.

---

## Round 0.2 · Minor Revisions

We have received a few minor suggestions from Reviewer #1. You are required to address those and resubmit.

Reviewer 1 ·

Basic reporting

The authors answered the queries.
They have provided the relevant updates.

Experimental design

The authors are using the terms as:
logPropabilityPosteriorWeights - in equation 5,6,7 etc.

They must check back and provide the correct terms - Example: logProbabilityPosteriorWeights
All the equations and the corresponding checks should be completed properly.

Validity of the findings

The experimental findings are updated as per the previous review.

Additional comments

The authors are required to update the above-stated items.
After the complete check and thorough update of the English, the article will be eligible to be published in the journal.

---

## Round 0.3 · accepted · Accept

All the suggested changes have been made. The manuscript may be accepted for publication in its current form.